# Learning Dense Flow Field for Highly-accurate Cross-view Camera Localization

**Zhenbo Song[1]***  **Xianghui Ze[1]***  **Jianfeng Lu[1]**  **Yujiao Shi[2]†**

[1]Nanjing University of Science and Technology, [2]ShanghaiTech University

{songzb, zexh, lujf}@njust.edu.cn, shiyj2@shanghaitech.edu.cn

## Abstract

This paper addresses the problem of estimating the 3-DoF camera pose for a ground-level image with respect to a satellite image that encompasses the local surroundings. We propose a novel end-to-end approach that leverages the learning of dense pixel-wise flow fields in pairs of ground and satellite images to calculate the camera pose. Our approach differs from existing methods by constructing the feature metric at the pixel level, enabling full-image supervision for learning distinctive geometric configurations and visual appearances across views. Specifically, our method employs two distinct convolution networks for ground and satellite feature extraction. Then, we project the ground feature map to the bird's eye view (BEV) using a fixed camera height assumption to achieve preliminary geometric alignment. To further establish the content association between the BEV and satellite features, we introduce a residual convolution block to refine the projected BEV feature. Optical flow estimation is performed on the refined BEV feature map and the satellite feature map using flow decoder networks based on RAFT. After obtaining dense flow correspondences, we apply the least square method to filter matching inliers and regress the ground camera pose. Extensive experiments demonstrate significant improvements compared to state-of-the-art methods. Notably, our approach reduces the median localization error by 89%, 19%, 80%, and 35% on the KITTI, Ford multi-AV, VIGOR, and Oxford RobotCar datasets, respectively.

## 1 Introduction

Cross-view camera localization, also known as ground-to-aerial/satellite image matching, has evolved from the study of coarse localization through image retrieval[4, 12, 16, 20] to a highly accurate 3-DoF pose estimation task[5, 13, 22, 24]. This task involves accurately determining the precise position and orientation of a ground camera in relation to a satellite patch that encompasses the surrounding environment.

The key challenge in cross-view camera localization is to bridge the gap between the visual observations captured by the ground camera and the corresponding top-view camera, enabling robust and accurate regression of the camera's pose in real-world spatial coordinates. This capability holds immense significance across various applications, including augmented reality, autonomous navigation, virtual reality, and robotics, where precise camera localization is crucial for enabling advanced functionalities and interactions with the environment.

This paper focuses on investigating the problem of single-frame ground image localization, which aims to estimate the 3 DoF pose of a ground camera relative to a single satellite image. This problem

---

*Equal contribution

†Corresponding author

37th Conference on Neural Information Processing Systems (NeurIPS 2023).

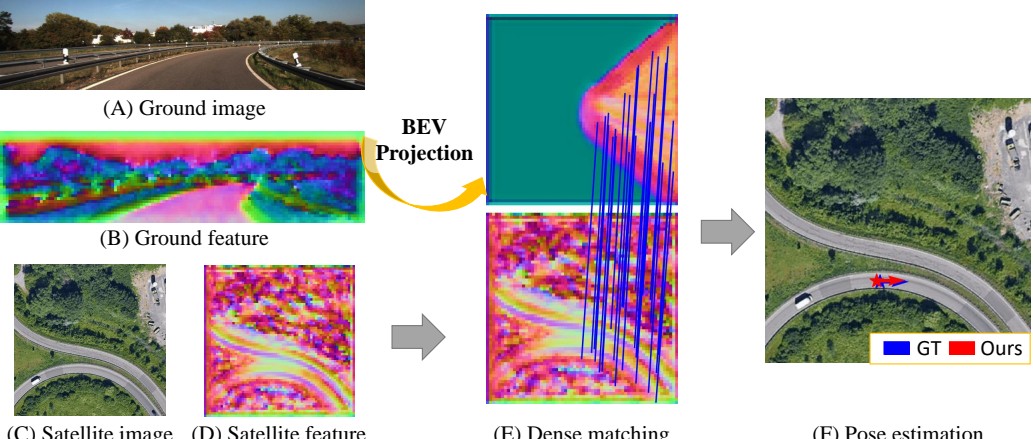

(A) Ground image

BEV Projection

(B) Ground feature

(C) Satellite image (D) Satellite feature      (E) Dense matching      (F) Pose estimation

GT   Ours

Figure 1: Visualizations of the ground and aerial feature maps, and the dense matching correspondences cross views for localization. For visual simplicity, dense matching displays only partial inliers.

assumes that the satellite image covers the same geographic region as the ground image. The objective of localization is to determine which pixel of the satellite image corresponds to the ground camera's position and its current shooting direction[24]. In practical applications, as each pixel of the satellite image is associated with GPS labels, it becomes feasible to obtain the geographical location information and spatial orientation information of the ground camera.

Most current methods address this problem by implicitly learning cross-view correspondences to regress camera localization [5, 13, 22, 24]. These approaches establish aggregated descriptor-based representations for both ground and aerial images, resulting in relatively short inference time. However, these methods suffer from two main issues. Firstly, they lack an explicit establishment of spatial unification between views and heavily rely on brute-force training to simultaneously solve spatial and semantic alignment. Secondly, in theory, precise relative pose estimation requires obtaining local cross-view matching relationships, such as point-to-point correspondences. Implicit cross-view feature correlation and pose regression may lead to a loss in localization accuracy. In contrast, methods[13] based on point-to-point matches have been proposed, which utilize LM optimization for feature map alignment between the ground and satellite views. These methods often leverage end-to-end ground truth pose as supervision to learn local correspondences between cross-views. Such supervision may be weak for point-wise deep feature metric learning. Based on the above observations, we propose a method that explicitly represents point-wise correspondences between cross-views, thus facilitating accurate pose estimation. Our key idea is to learn dense flow fields between cross-views and directly compute accurate ground camera poses from such flows in an end-to-end manner. To achieve this, we develop two distinct convolutional neural networks for feature extraction from the aerial and ground images, respectively. But ground and aerial feature maps exhibit significant domain differences in terms of spatial configuration and visual appearance, posing a critical challenge for the neural network to learn effective matching on such feature maps. Furthermore, due to the absence of depth information in the ground images, direct cross-view matching leads to a one-to-many correspondence issue. For instance, a building facade in the ground image may correspond to multiple edges in the aerial image. Existing methods often assume a fixed ground height to enable the projection from the ground to the satellite image[13]. In line with this approach, our method employs the same strategy to geometrically project the ground features, obtaining a bird's eye view (BEV) feature map that corresponds to the ground image. This projection facilitates spatial alignment between the cross-view feature maps. As mentioned above, for the uncertainty in the ground height assumption, we introduce a RefineBlock to further refine the BEV features and achieve content alignment. Subsequently, optical flow estimation is performed on the refined BEV feature map and the satellite feature map using flow decoder networks based on RAFT[17]. Specifically, we calculate multi-scale correlation volumes using the BEV and satellite features and employ GRU modules iteratively for dense flow estimation. Furthermore, we assign scores to each established matching relationship based on their similarity, effectively reducing the weight of erroneous matches. A notable advantage of computing flows on the BEV and satellite features is that the matching relationship allows us to directly utilize the least square method to solve the 3 DoF pose, *i.e.*, the

2-DoF translation, and 1-DoF rotation. In order to learn robust matching results, we use the dense optical flow as well as the ground truth pose as supervision to train the network progressively.

Our main contributions can be summarized as follows:

- This paper introduces a novel approach that learns dense pixel-wise flow fields between ground and satellite images. This enables accurate cross-view camera localization by directly calculating precise ground camera poses from the learned flow.

- We propose spatial alignment and content refinement modules for ground image feature enhancement. By leveraging the BEV representation of ground feature maps, the proposed method achieves accurate optical flow estimation according to both spatial and semantical alignments.

- We conduct extensive experiments on various datasets, including KITTI, Ford multi-AV, VIGOR, and Oxford RobotCar, which demonstrates that our method achieves superior performance and exhibits strong generalization ability across all datasets.

## 2 Related Works

**Localization based on retrieval.** Image retrieval methods are widely utilized for same-view localization tasks[2, 10–12]. This approach assumes the availability of a reference dataset containing a vast number of geotagged ground images. Query images are then matched with ground images in the reference dataset to determine their corresponding geographic locations. However, this method requires a significant number of ground markers, which can be expensive. As satellite maps become more widely available, cross-view localization has gained attention as a potential alternative. In cross-view retrieval tasks, a common approach is to create separate global descriptors for the ground image and the satellite image, and then compare their similarity[4, 14, 18, 19, 23]. Alternatively, recent studies have explored fusing consecutive multi-frame ground images to generate a descriptor, which is then matched with the descriptor of the satellite image[16, 20]. In this cross-view retrieval methods, the camera is assumed to be located at the center of the satellite map. However, this assumption limits the generalization of these methods.

**Cross-view camera pose estimation.** This work differs from the retrieval task in that it relies on an initial estimate of a given camera pose and then proceeds to refine it to obtain a pose that is close to the true value. Previous works have used various approaches to tackle this problem. For example, VIGOR[24] first uses an end-to-end framework to roughly locate the query through retrieval and then refines the location by predicting offsets through regression. Xia *et al.*[22] formulates the localization problem as a multi-class classification problem and uses a dense multimodal space for localization. [13] uses a geometric projection module to align the cross-view features, and then optimize the camera pose end-to-end using the Levenberg-Marquardt algorithm[6]. Wang *et al.*[21] uses 3D points to bridge the geometric gap between ground and overhead views, iteratively fusing ground images with 3D points iteratively for pose estimation. SliceMatch[5] is given a set of candidate photo poses and then compares a single ground descriptor with a set of pose-dependent satellite map descriptors to generate the final camera pose. It is worth noting that previous works have primarily relied on global descriptors [22, 24] or local descriptors[5, 6, 13, 21] for cross-view positional estimation. However, in our work, we propose an alternative approach that relies on dense point matching, which we believe to be a reliable basis for cross-view localization.

## 3 Methods

As shown in Figure 2, the proposed method is built upon the RAFT architecture. The main differences are the heterogeneous ground and aerial feature extractors and feature transformation modules, *i.e.* the ground feature projection module, and the BEV feature refinement module. In addition, we propose a differentiable least squares pose regression module to filter matching inliers and calculate the camera pose through dense flow fields. Mathematically, given a ground image $\mathbf{I}_g$ and a corresponding satellite image $\mathbf{I}_s$, the task is to determine the position $\boldsymbol{t} = (\Delta u, \Delta v)$ and orientation $\theta$ of the camera with respect to the satellite image. To achieve this, we propose a robust method that leverages the similarity between the points in the two images to estimate the corresponding satellite image $\hat{\boldsymbol{p}}$ coordinates of each point $\boldsymbol{p}$ in $\mathbf{I}_g$. By finding the position relation between points, we can derive a geometric relation

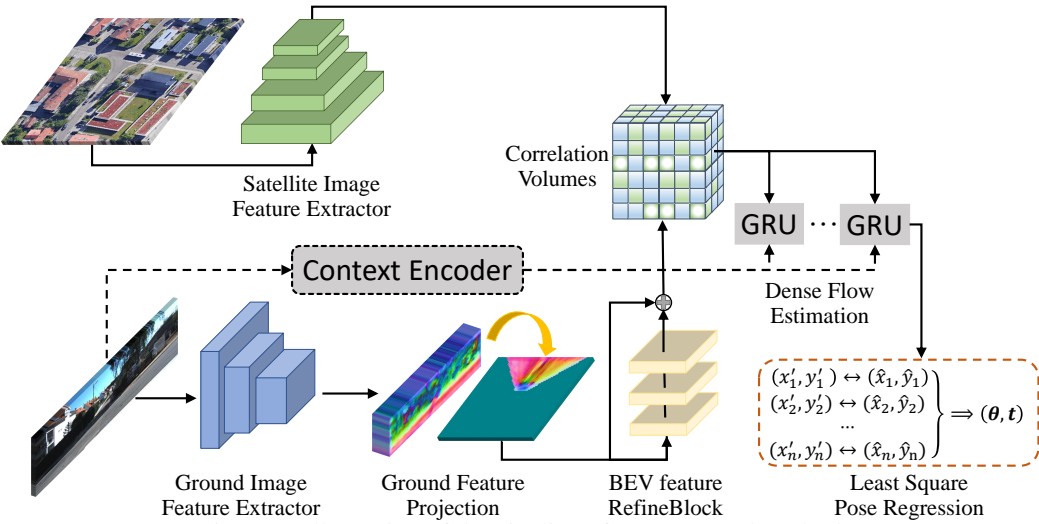

Figure 2: Illustration of the pipeline of our proposed method.

function $M$ that maps points between the two images such that $\hat{\boldsymbol{p}} = M(\boldsymbol{p})$. Next, we introduce the three key modules of our approach: (i) Feature extraction and view unification, (ii) Dense flow estimation, (iii) Least squares pose regression.

## 3.1 Feature Extraction and View Unification

In the presence of a large field-of-view error, it is a great challenge to match points directly to $\mathbf{I}_g$ and $\mathbf{I}_s$. Our work addresses the significant differences in field-of-view between ground and satellite images by introducing a feature extraction module that incorporates projection transformation information.

The input $\mathbf{I}_g$ and $\mathbf{I}_s$ are first mapped to the feature map, $\mathbf{F}_g = CNN_g(I_g)$ and $\mathbf{F}_s = CNN_s(I_s)$, where $CNN_g$ and $CNN_s$ are feature extraction modules. Merely relying on the feature extractor is insufficient to address the significant difference between the two fields of view. As a solution, we introduce the camera's intrinsics $\boldsymbol{K}$, extrinsics $\boldsymbol{T}$ to perform perspective transformation. Specifically, for any world point $\boldsymbol{x}^W$, the perspective transformation maps it to its corresponding image coordinate $\boldsymbol{x}^I$, as follows:

$$\boldsymbol{x}^I \simeq \boldsymbol{K}\boldsymbol{T}\boldsymbol{x}^W \tag{1}$$

Here, $\simeq$ denotes equality up to a scale factor. However, since there is no exact ground height, the correspondence between ground and satellite images is ambiguous. To address this issue, we exploit the fact that the overlap between ground and satellite images mainly occurs on the ground plane[13]. Therefore, within our ground feature projection module, we utilize Eq. 1 to map ground pixels $\boldsymbol{x}^I = [\boldsymbol{p}, 1]$ to the bird's-eye view (BEV) perspective via $\boldsymbol{p}' = G(\boldsymbol{p})$ by incorporating ground height $h$, where the world point is denoted by $\boldsymbol{x}^W = [\boldsymbol{p}', h, 1]$. Subsequently, we employ bilinear interpolation to sample the ground image and project it onto a top-down view using this correspondence.

To further enhance the quality of our BEV feature map, we further propose a $RefineBlock$ that addresses the following:

$$\mathbf{F}_{g2s} = RefineBlock(\mathbf{F}_g, \boldsymbol{p}') \tag{2}$$

The $RefineBlock$ is a key component in our network architecture, aiming to address the limitations of ground-to-overhead projection based on ground-plane homography, including incorrect correspondences for objects above the ground plane and distortions in the projections. To overcome these issues, the network should have a larger receptive field based on ground-plane homography projection of the input overhead view feature map for better and higher-level understanding of the feature map. Therefore, in the RefineBlock, a $7 \times 7$ convolutional layer is first employed to achieve a larger receptive field. Then, a $3 \times 3$ residual block is applied to refine the details in local regions. Finally, a $1 \times 1$ convolutional layer is utilized to facilitate feature interaction and integration, introducing enhanced non-linear transformation capabilities in the feature space. Ultimately, it generates a self-centered

BEV image that closely resembles the mental image humans construct during their own localization process.

## 3.2 Dense Flow Estimation

Similar to RAFT, the satellite features and the refined BEV features are subjected to a dot product operation, resulting in a four-dimensional correlation matrix that represents the similarity between every point in the two feature maps. This four-dimensional matrix is then subjected to average pooling with a $2 \times 2$ kernel on the last two dimensions, yielding features of varying resolutions. The features obtained from the different resolutions are then combined to form correlation volumes.

Our architecture consists of two separate feature extractors, namely the feature encoder and the context encoder. Here, we have utilized a feature encoder equipped with a view unification module to deal with the significant differences in viewing angles between the $\mathbf{I}_g$ and $\mathbf{I}_s$. The context encoder has the same architecture as the feature encoder but is designed to extract only semantic and contextual information from the reference image $\mathbf{I}_g$.

Finally, we create an index based on the initialized match positions to query the correlation volume. We use the GRU structure to iteratively update the match positions, which generates the final match results $p' \leftrightarrow \hat{p}$ as well as the match score $\mathbf{S}$ for each point, which is dependent on the similarity scores obtained from the matching process.

## 3.3 Least Squares Pose Regression

In the case of a BEV image $\mathbf{F}_{g2s}$ centered around itself, alignment with an uncalibrated satellite image can be achieved through rotation and translation. By doing so, it becomes apparent that the points $p'_i$ contained in $\mathbf{F}_{g2s}$ and their corresponding points $\hat{p}_i$ in $\mathbf{F}_s$ are related through Euclidean transformations. So our objective is to find the optimal values of the positional parameters $\theta$ and $t$ of the ground camera by minimizing the following loss function:

$$\xi = \arg\min_{\theta, t} \sum_{i=1}^{n} S_i (\boldsymbol{R}p'_i + t - \hat{p}_i)^2, \boldsymbol{R} = \begin{bmatrix} \cos\theta & -\sin\theta \\ \sin\theta & \cos\theta \end{bmatrix} \tag{3}$$

Here, the confidence score of the $i$-th match is represented by $S_i$, which is comprised of two components. The first component indicates whether the $i$-th match is visible in the BEV image, with a value of 1 for visible matches and 0 for invisible matches. The second component is the score of the $i$-th match, which is calculated by the network based on the similarity of the matches.

By using the least square methods, we can calculate the relative translation and rotation of the camera in the satellite image through matching pairs of points. Detailed computation procedure is presented in the appendix materials. It is noteworthy that this process is differentiable, which implies that we can compute gradients and optimize the model's parameters through backpropagation.

## 3.4 Training Objective

Our loss function $\mathcal{L}$ consists of three components: matching loss $\mathcal{L}_m$, confidence loss $\mathcal{L}_c$, and position loss $\mathcal{L}_p$.

**Matching Loss.** The process of mapping matching points between $\mathbf{F}_{g2s}$ and $\mathbf{F}_s$ is supervised by $\mathcal{L}_m$. To evaluate the accuracy of the matching points, we calculate the distance between the matching points and the real matching points.

$$\mathcal{L}_m = \sum_{i=1}^{n} \|f_i^{gt} - f_i^{pred}\|_1 \tag{4}$$

where for the $i$-th point, $f_i^{gt}$ denotes its ground truth flow and $f_i^{pred}$ denotes the predicted flow. The true matching points are generated by the camera poses, which provide a way to calculate the corresponding positions of each point in both $\mathbf{F}_{g2s}$ and $\mathbf{F}_s$.

**Confidence Loss.** The loss of confidence generated by the network is measured using $\mathcal{L}_c$. For each matching point, the network generates a confidence score, and we aim for high confidence when the network predicts a small distance between the matching point and the true matching point, and low confidence when it predicts a large distance.

$$\mathcal{L}_c = \sum_{i=1}^{n} \| \frac{S_i}{1 + exp(-\tilde{d}_i/\kappa)} + \frac{1 - S_i}{1 + exp(\tilde{d}_i/\kappa)} \|_1 \tag{5}$$

To balance the two terms of the formula, we first transform $d$ into a standard normal distribution by using the formula $\tilde{d}_i = (d_i - mean(d))/std(d)$. We then use $\kappa$ to make the results more discretely distributed. In the beginning of the experiment, the variance of $d$ is large, so we set $\kappa$ to 200. After fifteen rounds of training, $dis$ converges and the distribution becomes more concentrated. At this point, we set $\kappa$ to 20.

**Position Loss.** We also utilized end-to-end supervised prediction, using $\mathcal{L}_p$, to estimate the difference between the predicted location and the true location. Although $\mathcal{L}_m$ already incorporates location information, we observed that incorporating end-to-end supervision further improved the accuracy of the network's predictions.

$$\mathcal{L}_p = \|\theta_{gt} - \theta\|_1 + \|\boldsymbol{t}_{gt} - \boldsymbol{t}\|_1 \tag{6}$$

where $\theta_{gt}$ and $\boldsymbol{t}_{gt}$ is the GT camera pose.

**Total Loss.** The overall loss is the sum of the matching loss $\mathcal{L}_m^l$ and confidence loss $\mathcal{L}_c^l$ generated by each iteration, as well as the final resulting position loss $\mathcal{L}_p$.

$$\mathcal{L} = \beta\mathcal{L}_p + \sum_{i=1}^{l}(\mathcal{L}_m^l + \alpha\mathcal{L}_c^l) \tag{7}$$

Here, the subscript $l$ denotes the $l$-th iteration of the GRU. During training, we set $\alpha = 100$ and initialize $\beta = 1$. After 15 rounds of training, we increase the momentum parameter to $\beta = 10$. This adjustment is made to prevent the model from getting trapped in local optima during the initial stage of training.

## 4 Experiments

In this section, we will begin by introducing the dataset and evaluation metrics employed. After that, we conduct a comparison between our proposed methodology and state-of-the-art approaches. Finally, we will present a comprehensive ablative study.

### 4.1 Datasets

We conducted tests on the KITTI[3, 13], Ford multi-AV[1, 13], VIGOR[5, 24], and Oxford RobotCar[8, 9, 22] datasets. The query images in KITTI, Ford multi-AV, and Oxford RobotCar datasets have a limited horizontal field of view (HFoV), while the query images in the VIGOR dataset are panoramic. Given that KITTI and VIGOR were captured on different road segments, we evaluated the model's performance on both same-area and cross-area scenarios using these two datasets. Ford multi-AV was mainly captured in suburban areas with fewer reference points, and Oxford RobotCar was captured on the same road segment at different times. We tested the model's generalization ability to different environments, including suburban scenes, varying lighting conditions, and road situations, on these two datasets. The datasets used in this paper are obtained under academic licenses and are not original datasets specifically created for this work. A more detailed description of each dataset will be provided in the appendix.

### 4.2 Evaluation Metrics

We followed the settings of [5, 13, 22]. Specifically, we calculate the mean and median errors in meters between predicted and ground truth positions, as well as the mean and median angular errors

Table 1: Location and orientation estimation error and recall on KITTI dataset[3, 13]. Angular noise was constrained within a range of $\pm 10°$[13].

| Area | Model | Location(m) mean | median | Lateral(%) r@1m | r@5m | Longitudinal(%) r@1m | r@5m | Azimuth(°) mean | median | Azimuth(%) r@1° | r@5° |
|------|-------|----------|--------|------|------|------|------|------|--------|------|------|
| Same | DSM*[15] | - | - | 10.12 | 48.24 | 4.08 | 20.14 | - | - | 3.58 | 24.44 |
| | LM[13] | 12.08 | 11.42 | 35.54 | 80.36 | 5.22 | 26.13 | 3.72 | 2.83 | 19.64 | 71.72 |
| | SliceMatch[5] | 7.96 | 4.39 | 49.09 | 98.52 | 15.19 | 57.35 | 4.12 | 3.65 | 13.41 | 64.17 |
| | Ours | **1.48** | **0.47** | **95.47** | **99.79** | **87.89** | **94.78** | **0.49** | **0.30** | **89.40** | **99.31** |
| Cross | DSM*[15] | - | - | 10.77 | 48.24 | 3.87 | 19.50 | - | - | 3.53 | 23.95 |
| | LM[13] | 12.58 | 12.11 | 27.82 | 72.89 | 5.75 | 26.48 | 3.95 | 3.03 | 18.42 | 71.00 |
| | SliceMatch[5] | 13.50 | 9.77 | 32.43 | 86.44 | 8.30 | 35.57 | 4.20 | 6.61 | **46.82** | 46.82 |
| | Ours | **7.97** | **3.52** | **54.19** | **91.74** | **23.10** | **61.75** | **2.17** | **1.21** | 43.44 | **89.31** |

Table 2: Location and orientation estimation error and recall on Ford multi-AV dataset[1, 13]. angular noise was constrained within a range of $\pm 10°$[13].

| Area | Model | Location(m) mean | median | Lateral(%) r@1m | r@5m | Longitudinal(%) r@1m | r@5m | Azimuth(°) mean | median | Azimuth(%) r@1° | r@5° |
|------|-------|----------|--------|------|------|------|------|------|--------|------|------|
| Log1 | DSM*[15] | - | - | 12.00 | 53.67 | 4.33 | 21.43 | - | - | 3.52 | 13.33 |
| | LM[13] | 12.54 | 12.63 | 48.57 | 71.57 | **5.90** | **26.33** | 3.13 | 1.29 | 42.90 | 79.62 |
| | Ours | **10.55** | **10.19** | **68.67** | **95.48** | 5.48 | 25.86 | **1.43** | **0.47** | **75.76** | **94.29** |
| Log2 | DSM*[15] | - | - | 8.45 | 37.64 | 3.94 | 21.41 | - | - | 2.23 | 13.42 |
| | LM[13] | 12.01 | 11.49 | 29.94 | 77.78 | 4.94 | 26.00 | 4.36 | 3.75 | 15.35 | 63.00 |
| | Ours | **10.50** | **9.40** | **33.43** | **97.59** | **15.62** | **40.94** | **1.65** | **1.03** | **48.99** | **95.57** |

in degrees between predicted and ground truth bearings. For the KITTI and Ford multi-AV datasets, we also included the recall values for longitudinal and lateral localization errors, as well as orientation estimation errors, under certain thresholds. Our localization thresholds were set at 1 meter and 5 meters, while our orientation estimation thresholds were set at 1 degree and 5 degrees.

### 4.3 Implementation Details

We employed ResNet18 to extract features from both ground and satellite images, respectively, ensuring that the resolution was maintained at 1/8th of the original with 256 channels. Subsequently, in the dense flow estimation stage, we performed 12 iterations to obtain the final dense matching relationship. Finally, we used the least squares method to estimate relative position and relative angles based on the estimated matching relationships. During the training period, we utilized the Adam optimizer[7] for end-to-end training, with a learning rate of $2 \times 10^{-5}$, $\beta_1 = 0.9$, and $\beta_2 = 0.999$. The entire training schedule consisted of 25 epochs. At the 15th epoch, we adjusted the parameter $\kappa$ in the loss function from 200 to 20 and $\beta$ from 1 to 10. We conducted 25 rounds of training on two TITAN V GPUs with a batch size of 6. Each epoch required approximately two hours of training time. During the evaluation process, estimating the pose of a ground image took an average of 0.25 seconds and approximately occupied 6GB of GPU memory.

### 4.4 Same-Area Evaluations

We conducted performance evaluations of the model in the same region on both the KITTI and Vigor datasets. The results of the KITTI dataset tests are shown in Table 1. Our method significantly outperformed previous algorithms in camera pose estimation, with an average error reduction of 81% and a median error reduction of 89% compared to SliceMatch. Notably, our algorithm's predictive performance for vertical position in the same region was significantly higher than that of other algorithms, indicating that despite the limited field of view, our approach learned the semantic relationship between roadside buildings in this area and their corresponding features in satellite images.

Table 3: Location and orientation estimation error and recall on VIGOR dataset[5, 24]. Aligned Images means the ground image orientation is known.

| Model | Aligned | Same-Area | | | | Cross-Area | | | |
| | | Location(m) | | Azimuth(°) | | Location(m) | | Azimuth(°) | |
| | | mean | median | mean | median | mean | median | mean | median |
|---|---|---|---|---|---|---|---|---|---|
| CVR[24] | √ | 8.99 | 7.81 | - | - | 8.89 | 7.73 | - | - |
| MCC[22] | √ | 6.94 | 3.64 | - | - | 9.05 | 5.14 | - | - |
| SliceMatch[5] | √ | 5.18 | 2.58 | - | - | 5.53 | 2.55 | - | - |
| Ours | √ | **3.03** | **0.97** | - | - | **5.01** | **2.42** | - | - |
| SliceMatch[5] | × | 8.41 | 5.07 | 28.43 | 5.15 | 8.48 | 5.64 | 26.20 | 5.18 |
| Ours | × | **4.97** | **1.90** | **11.20** | **1.59** | **7.67** | **3.67** | **17.63** | **2.94** |

Table 4: Location error on Oxford RobotCar dataset[8, 9, 22]. The orientation of the ground image is known[22].

| Model | Test1 | | Test2 | | Test3 | |
| | Location(m) | | Location(m) | | Location(m) | |
| | mean | median | mean | median | mean | median |
|---|---|---|---|---|---|---|
| CVR[24] | 1.88 | 1.47 | 2.64 | 1.99 | 2.35 | 1.71 |
| MCC[22] | 1.42 | 1.10 | 1.95 | 1.33 | 1.94 | 1.29 |
| Ours | **1.17** | **0.72** | **1.76** | **0.97** | **1.79** | **0.92** |

In Table 3, the testing results on the VIGOR dataset show that our method achieves significantly lower average and median localization errors than SliceMatch by 42% and 62%, respectively, when the direction is known. When the direction is unknown, our algorithm still performs well, with both the average and median localization errors lower than those of SliceMatch.

Furthermore, our algorithm demonstrates good performance in angle prediction, with average and median angle errors lower than those of SliceMatch by 67% and 69%, respectively.

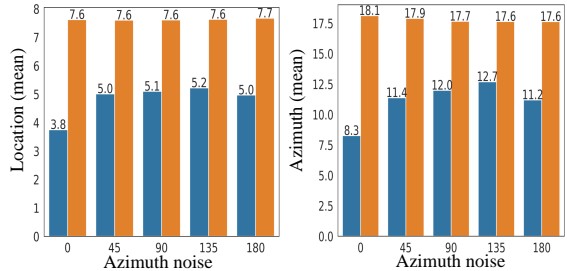

Figure 3: The localization error under varying noise conditions at different angles on the VIGOR dataset. Orange indicates cross-area, and blue indicates same-area.

## 4.5 Cross-Area Generalization

Generalizing new ground images across different regions is a more challenging task than within the same region, as the test area may appear vastly different from the training area. Despite this increased difficulty, our approach still demonstrates strong performance. As shown in Table 1 Cross-Area, Our average localization error is 40% lower than SliceMatch, and our median localization error is 64% lower than SliceMatch. Compared to descriptor-based matching methods, our approach exhibits greater robustness.

In the cross area of VIGOR, our algorithm performs better than previous methods in both known and unknown directions. Additionally, as shown in Figure 3, we tested the performance of our algorithm across various angles using a model trained in unknown directions. Our algorithm demonstrated strong stability against noise for different angle variations in both the same area and cross area.

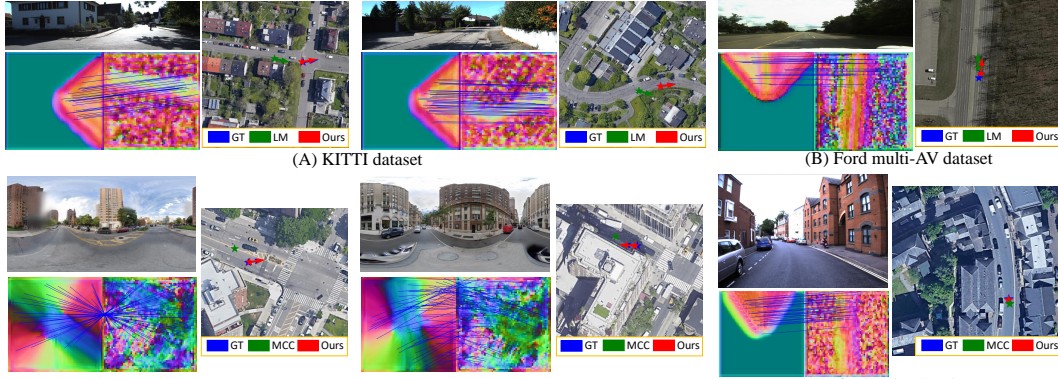

Figure 4: For each scene, the up left is the ground image, the bottom left denotes matching inliers and the right shows the satellite image and localization results.

Table 5: Ablation Study on the KITTI dataset[3, 13]

| Model | Same-Area | | | | Cross-Area | | | |
|---|---|---|---|---|---|---|---|---|
| | Location(m) | | Azimuth(°) | | Location(m) | | Azimuth(°) | |
| | mean | median | mean | median | mean | median | mean | median |
| Ours w/o Ground Feature Projection | 3.52 | 2.66 | 2.09 | 1.48 | 9.83 | 7.11 | 3.13 | 2.22 |
| Ours w/o BEV Feature RefineBlock | 2.02 | 0.78 | 0.82 | 0.53 | 8.72 | 4.55 | 2.56 | 1.47 |
| Ours w/o Matching Loss | 9.06 | 8.10 | 1.93 | 1.12 | 10.60 | 9.97 | 2.75 | 1.82 |
| Ours w/o Position Loss | **0.99** | 0.53 | 0.51 | 0.36 | 9.63 | 4.79 | 2.87 | 1.65 |
| Ours w/o Confidence | 1.65 | 0.65 | 0.98 | 0.75 | **7.76** | 3.57 | 2.31 | 1.50 |
| Ours | 1.48 | **0.47** | **0.49** | **0.30** | 7.97 | **3.52** | **2.17** | **1.21** |

## 4.6 Vary-Environment Generalization

In addition, to validate the performance of our algorithm under different environmental conditions, we tested it on the Ford multi-AV dataset and the Oxford RobotCar dataset. The results of the Ford multi-AV dataset are presented in Table 2, which is primarily focused on suburban scenes with fewer surrounding buildings and difficulty in obtaining semantic information, resulting in generally poor localization accuracy. Moreover, in the limited HFOV scenario, the longitudinal position is more challenging than the lateral position in the driving direction. Consequently, the recall rate for the longitudinal position is significantly lower than that of the lateral position, and this trend applies to all the compared methods. Nonetheless, our algorithm still outperforms previous algorithms on this dataset, and we found that our lateral recall rate is significantly higher, indicating our algorithm's ability to capture road surface information under semantically unclear conditions. Our algorithm demonstrated superior performance across the three distinct test sets of the Oxford RobotCar dataset, as shown in Table 4. These results suggest that our approach exhibits strong generalization capabilities across diverse environmental conditions, seasonal variations, and roadway scenarios.

## 4.7 Ablation Study

Table 5 presents the results of our ablation experiments conducted on the KITTI dataset to evaluate the efficacy of the view unification mechanism comprised of Ground Feature Projection and BEV Feature RefineBlock, the loss function composed of Matching Loss and Position Loss, and the Confidence mechanism. We employ Ground Feature Projection and BEV Feature RefineBlock to address significant viewpoint differences between cross-view images and incorporate BEV and satellite image features within the same scope. Our experiments demonstrate that utilizing Ground Feature Projection and BEV Feature RefineBlock significantly improves matching accuracy, reducing average localization error by 57% and 27%, respectively. Furthermore, by incorporating matching points during the training process and employing a more robust supervised training approach, we

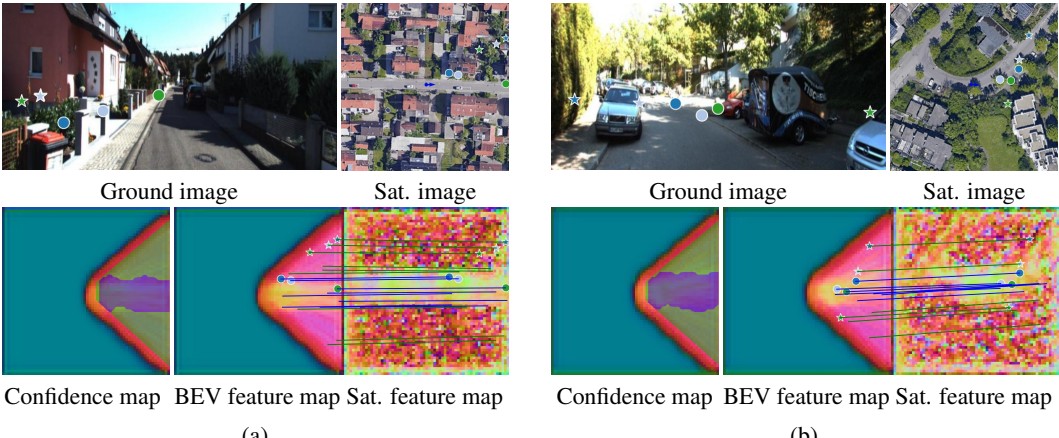

| Ground image | Sat. image | | Ground image | Sat. image |

| Confidence map | BEV feature map | Sat. feature map | Confidence map | BEV feature map | Sat. feature map |

(a) (b)

Figure 5: Matching confidence illustration. The top row of each subfigure shows the ground and satellite images, highlighting corresponding points. Circles indicate confident correspondences (score > 0.5), while stars represent potentially incorrect correspondences (score < 0.5). Matching points share the same color and shape in both images. In the bottom row, the left image shows confidence scores on the BEV feature map (blue for > 0.5, green for < 0.5). The middle and right images depict the BEV and satellite feature maps, respectively, with connected lines indicating matching pixels. The circles and stars in these images align with the corresponding points in the top row.

observe a 57% decrease in average localization error. Finally, the addition of a confidence mechanism further enhances network precision, resulting in a 10% reduction in average error. Through conducting ablation experiments, we have demonstrated the efficacy of the view unification, loss, and confidence mechanisms proposed by us.

In Figure 5, we conducted a qualitative analysis of the confidence mechanism on the KITTI dataset. We utilized a threshold of 0.5 to distinguish between high-confidence matches and low-confidence matches. In these visualizations, circles represent matching points with high confidence, while stars indicate matching points with low confidence. Additionally, for clarity, we displayed estimated corresponding points on both the original ground and satellite images. From the visualizations, it is evident that pixels in the bird's-eye-view image, which are mutually visible in both ground and satellite images (e.g., road areas), tend to have higher confidence. On the other hand, pixels that are visible only in one view (such as building rooftops, tree canopies, and occluded regions) generally exhibit lower confidence. As a result, our confidence map effectively selects reliable matching points for accurate pose estimation. This holds significant implications for improving precision.

# 5 Conclusion

This paper presents a novel approach based on dense matching to address the problem of cross-view localization. Our approach utilizes a view unification module to bridge the visual gap between satellite and ground images. Additionally, we introduce a matching loss to provide stronger supervision for the model during training. To evaluate the effectiveness of our algorithm, we conduct experiments on four datasets that vary in geographical regions, environmental conditions, and road scenarios. Our results demonstrate the impressive generalization capability of our approach. One possible future direction is to explore the integration of spatial and temporal information, to further improve the accuracy and robustness of our approach for cross-view localization.

# 6 Acknowledgement

This work was supported in part by the National Natural Science Foundation of China (No. 62302220), in part by the China Postdoctoral Foundation (No. 2023M731691), and in part by the Jiangsu Funding Program for Excellent Postdoctoral Talent (No.2022ZB268).

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

# Appendix

## A  Theoretical Proof

We provide a detailed derivation and proof of the least-squares pose regression process in the text. The camera position parameters $\theta$ and $t$ can be obtained by solving the following loss function.

$$\xi = \arg\min_{\theta,t} \sum_{i=1}^{n} S_i(\boldsymbol{R}\boldsymbol{p}'_i + \boldsymbol{t} - \hat{\boldsymbol{p}}_i)^2, \boldsymbol{R} = \begin{bmatrix} \cos\theta & -\sin\theta \\ \sin\theta & \cos\theta \end{bmatrix} \quad (8)$$

The relationship between the corresponding points $\boldsymbol{p}'_i$ and $\hat{\boldsymbol{p}}_i$ in point sets $\boldsymbol{p}'$ and $\hat{\boldsymbol{p}}$ can be expressed as $\boldsymbol{p}'_i = \boldsymbol{R}\hat{\boldsymbol{p}}_i + \boldsymbol{t}$. To simplify the problem, we calculate the centroids $\boldsymbol{g}'$ and $\hat{\boldsymbol{g}}$ of point sets $\boldsymbol{p}'$ and $\hat{\boldsymbol{p}}$, respectively. We then subtract the centroids from each point in their respective sets, resulting in $\boldsymbol{q}'_i = \boldsymbol{p}'_i - \boldsymbol{g}'$ and $\hat{\boldsymbol{q}}_i = \hat{\boldsymbol{p}}_i - \hat{\boldsymbol{g}}$. This simplifies the problem to:

$$\begin{aligned} \xi &= \arg\min_{\theta} \sum_{i=1}^{n} S_i(\boldsymbol{R}\boldsymbol{q}'_i - \hat{\boldsymbol{q}}_i)^2 \\ &= \arg\min_{\theta} \sum_{i=1}^{n} S_i(\boldsymbol{R}\boldsymbol{q}'_i - \hat{\boldsymbol{q}}_i)^t(\boldsymbol{R}\boldsymbol{q}'_i - \hat{\boldsymbol{q}}_i) \\ &= \arg\min_{\theta} \sum_{i=1}^{n} S_i({\boldsymbol{q}'_i}^t\boldsymbol{q}'_i) + \hat{\boldsymbol{q}}_i^t\hat{\boldsymbol{q}}_i - 2\hat{\boldsymbol{q}}_i^t\boldsymbol{R}\boldsymbol{q}'_i \end{aligned} \quad (9)$$

Therefore, minimizing $\xi$ is equivalent to maximizing $\sum_{i=1}^{n} S_i(\hat{\boldsymbol{q}}_i^t\boldsymbol{R}\boldsymbol{q}'_i)$. Assuming that $\boldsymbol{H} = \sum_{i=1}^{n} S_i(\boldsymbol{q}'_i\hat{\boldsymbol{q}}_i^t)$. then

$$\sum_{i=1}^{n} S_i(\hat{\boldsymbol{q}}_i^t\boldsymbol{R}\boldsymbol{q}'_i) = tr(\sum_{i=1}^{n} \boldsymbol{R}S_i\boldsymbol{q}'_i\hat{\boldsymbol{q}}_i^t) = tr(\boldsymbol{R}\boldsymbol{H}) \quad (10)$$

Upon performing an SVD decomposition on $\boldsymbol{H}$, the result is $\boldsymbol{H} = \boldsymbol{U}\boldsymbol{\Lambda}\boldsymbol{V}^t$, where $\boldsymbol{U}$ and $\boldsymbol{V}$ are orthogonal matrices of dimensions $3 \times 3$, and $\boldsymbol{\Lambda}$ is $3 \times 3$ diagonal matrix with non-negative elements.

Then we introduce an orthogonal matrix $\boldsymbol{X} = \boldsymbol{U}\boldsymbol{V}^t$, and observe that $\boldsymbol{X}\boldsymbol{H} = \boldsymbol{V}\boldsymbol{\Lambda}\boldsymbol{V}^t$ is a symmetric positive-definite matrix. According to the properties of positive definite matrices, it holds that $tr(\boldsymbol{X}\boldsymbol{H}) \geq tr(\boldsymbol{B}\boldsymbol{X}\boldsymbol{H})$ for any $3 \times 3$ orthogonal matrix $\boldsymbol{B}$. Therefore, we obtain the maximum value of $\sum_{i=1}^{n} S_i(\hat{\boldsymbol{q}}_i^t\boldsymbol{R}\boldsymbol{q}'_i)$ when $\boldsymbol{R} = \boldsymbol{X}$, which implies that

$$\boldsymbol{R} = \boldsymbol{U}\boldsymbol{V}^t \quad (11)$$

The dense matching relationship between the graphs gives rise to the uniqueness of solution for R. Subsequently, the relative displacement can be determined as

$$\boldsymbol{t} = \boldsymbol{g}' - \boldsymbol{R}\hat{\boldsymbol{g}} \quad (12)$$

By utilizing equations 11 and 12, we can obtain the camera location and azimuth through a straight-forward differentiable process, using the least-squares regression method.

## B  Datasets

**KITTI.**  The ground images in the KITTI dataset were obtained from the original data. [13] collected satellite images of the KITTI dataset, with each ground image accompanied by a corresponding satellite image whose center corresponds to the position of the ground camera. The dataset is divided into Training, Test1, and Test2 subsets. The images in Test1 are from the same area as the images in the training set, while the images in Test2 are from different areas. The ground resolution of the satellite images in this dataset is approximately 0.2 meters per pixel, and the coverage area of the satellite images is approximately $100 \times 100$ square meters.

**Ford Multi AV.**  For the Ford Multi AV dataset, we only utilized the data of the V2 vehicles from the original dataset. Shi and Li first plotted the trajectories of the ground-view images and collected satellite images along these trajectories. Ground images captured on one date were used for training, while images captured on another date were used for testing. The satellite images remained unchanged during both training and testing. The ground resolution of the satellite images in this dataset is approximately 0.2 meters per pixel, and the coverage area of the satellite images is approximately $100 \times 100$ square meters.

**VIGOR.** The satellite images of the VIGOR dataset were collected by [24]. The ground images are north-aligned panoramas, and they are from four cities: New York, San Francisco, Chicago, and Seattle. There is a positive satellite image in the database for each ground image, meaning that the ground image is within the center 1/4 quarter of the satellite image. All the satellite images are north-aligned. The dataset includes two evaluation splits: same-area and cross-area, based on whether the images in training and testing sets are from the same region. Research is conducted to estimate the relative pose between the ground image and its positive satellite image center. In this dataset, the ground resolution of satellite images for New York, San Francisco, Chicago, and Seattle is 0.113, 0.118, 0.111, and 0.101, respectively. The size of the aerial views for these four cities is $640 \times 640$.

**Oxford RobotCar.** The satellite images of the Oxford RobotCar dataset were collected by [8]. They provided a very large satellite map covering the whole region where the ground images were collected. The ground images were collected through multiple traversals on a single route in Oxford, UK, encompassing various time periods, seasons, and weather conditions. During implementation, for each ground image, a random relative translation is generated, and a small satellite patch corresponding to the randomly generated pose is extracted from the large satellite map. Research is conducted to estimate the relative pose. The ground sample distance of the satellite images in this dataset is 0.0924m per pixel, and the coverage of the satellite images used is around $55 \times 55$ square meters.

## C  Comparison with LM[13]

To make a fair comparison with the LM[13], we trained both the LM and our method for an equal number of epochs and found that our method still outperforms the LM method.

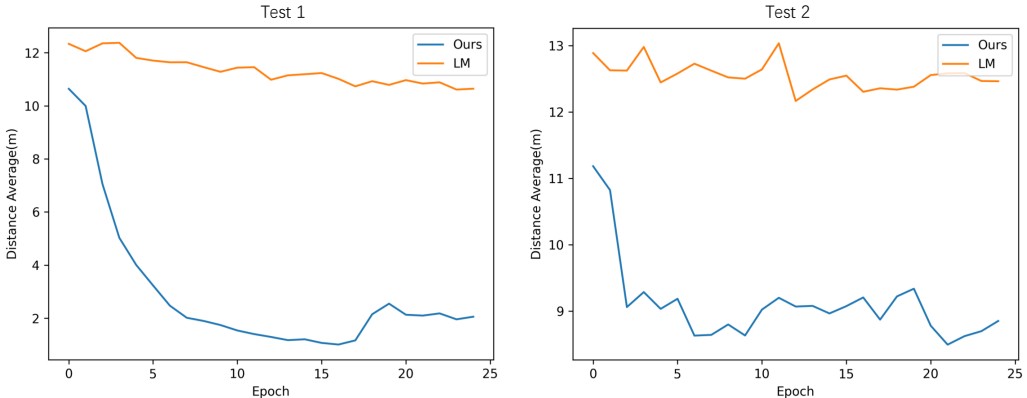

Figure 6: Performance comparison between LM[13] and our method with longer training epochs on the KITTI dataset.

The LM[13] aims to optimize the relative pose by minimizing the feature differences between corresponding pixels in two-view images. However, the pixel correspondences are computed based on planar homography (and relative camera pose), which is incorrect for objects above the ground plane. Minimizing feature differences between non-corresponding features leads to suboptimal relative pose estimation. In contrast, our objective in this paper is to first estimate the precise correspondences between the two views and then compute the relative pose. The estimation of correspondences is independent of the relative pose. Once the correspondences are sufficiently accurate and robust, the relative pose can be directly computed by solving a least squares problem instead of training additional networks. Therefore, the main capability of our method lies in accurate correspondence learning rather than relative pose estimation. As a result, our method achieves better performance.

