# OpenReview forum: "Learning Dense Flow Field for Highly-accurate Cross-view Camera Localization"
_NeurIPS.cc/2023/Conference — NeurIPS 2023 poster_

### Official Review · Reviewer_NkYA · 2023-06-30

**Soundness:** 3 good
**Presentation:** 3 good
**Contribution:** 2 fair
**Rating:** 5
**Confidence:** 3

**Summary:**

The paper addresses the problem of cross-view camera localization, i.e. of estimating a 3D camera pose for a ground-level image with respect to a satellite image of its local surroundings (assuming a coarse localization is already given). For this, it proposes a deep learning pipeline that aims at estimating a dense flow field between the jointly computed feature maps of the satellite image and a bird's-eye-view (BEV) rendering of the ground-level feature maps. Based on an initial correlation volume between those two feature maps, it then employs several GRU stages (similar to RAFT) in order to estimate the dense flow field and finally performs least-squares pose regression. The approach is quantitatively evaluated on the KITTI, Ford multi-AV, VIGOR and Oxford RobotCar datasets. Compared to previous approaches, the proposed method achieves a considerably improved localization accuracy.


**Strengths:**

+ The proposed approach is well-engineered and achieves considerably improved localization accuracies on all four datasets, compared with the previous SOTA methods.

+ Experimental evaluations of 4 datasets and comparisons against several published baselines support the claimed contributions.

+ The effects of the different elements of the proposed approach are investigated in an ablation study.


**Weaknesses:**

- The paper makes a relatively minor technical contribution. It presents a novel network architecture for a very specialized problem setting. The architecture is well suited for the estimation problem at hand and achieves very good localization accuracy, but the impact will still be limited, since the problem setting is interesting only for a relatively small subcommunity. Specifically, there is no strong machine learning content that would make this a particularly good fit for NeurIPS.

- As the proposed approach relies on a coarse localization being already given, its success in practice will heavily rely on how robust the estimation is with respect to a coarse localization error. In other words, how close will the location of the satellite image already have to be to the camera location and over what search range will the approach reliably find a correct localization. Unfortunately, this is not discussed at all, nor is it reflected in the experimental design. The quantitative evaluation follows "the settings of [14, 29, 43]", which presumably means that the ground truth location lies within the center quarter of the satellite image tile?


**Questions:**

This paper leaves a mixed impression. On the one hand, the proposed approach is well-engineered and achieves very good results. On the other hand, I am also not terribly excited about the paper, since it addresses a rather niche topic without a strong learning-focused contribution. I would be more excited about it in a more application focused conference, but I do not think that it has something great to offer particularly for the NeurIPS audience.

**Limitations:**

.

---

> ### Author Rebuttal · Authors · 2023-08-10
>
> We thank Reviewer-NkYA for the valuable comments. Please find our response below.
>
>
> **1:Minor technical contribution, limited impact.**
>
> Instead of a trivial ensemble of existing techniques, our contribution is mainly on how to solve unique challenges in a new scenario faced by existing techniques.
>
> It is well-known that RAFT is a powerful network to estimate optical flow between two images with GT supervision, but it is unknown how to supervise RAFT when such GT supervision is unavailable, and the application scenario (ground-view Vs. overhead-view) significantly differs from the publicly available training dataset (ground-view Vs. ground-view). Furthermore, the extreme stereo case (ground Vs. overhead) makes the image reconstruction loss for self-supervision of the optical flow impossible due to severe occlusions and different capture times (different years, illumination, color, etc.)
>
>
> This paper discovers that the supervision is computable when converting the ground view to the overhead view by a BEV-synthesis, and introduces additional implicit supervisions either to strengthen the supervision (i.e., pose supervision in Eq. 6) or to filter out unreliable correspondences (i.e., confidence map with implicit supervision from pose error in Eq. 6 and flow error in Eq. 5.). Please refer to our response to the first question of Reviewer-Bgje for more details. The experiments in Tab. 5 also demonstrate that naively applying existing techniques for solving the task achieves undesirable performance. Only when incorporating our proposed solutions for addressing the unique challenges can the method achieve the best performance. Our improvement over the state-of-the-art is also significant.
>
> We believe this paper should also benefit some similar tasks. For example, the self-supervision in Eq. 5 for the confidence map is also applicable to other self-supervised optical flow estimation tasks.
>
>
>
> **2: Error range of the coarse locations.**
>
> The coarse locations of ground cameras on the VIGOR dataset are in the center quarter of the satellite image. The search range is approximately $36\times 36$ m$^2$. The location search range for the KITTI, Ford multi-AV, and Oxford RobotCar datasets is about $40 \times 40$ m$^2$. The error of the coarse pose in the experiment settings is similar to a consumer-level noisy GPS.
>
> We present the performance of our method across a broader location search range in Tab. C of the rebuttal PDF, specifically, $60 \times 60$ m$^2$. As the location search range increases, so does the location ambiguity, leading to a slight decline in our method's performance. However, we should note that even with the larger location search range of $60 \times 60$ m$^2$, our approach outperforms the recent state-of-the-art SliceMatch [6] when it is tested within a smaller location search range of $40 \times 40$ m$^2$.

---

> > ### Comment · Reviewer_NkYA · 2023-08-21
> >
> > Thank you for those clarifications.

---

### Official Review · Reviewer_63Cb · 2023-07-03

**Soundness:** 2 fair
**Presentation:** 3 good
**Contribution:** 1 poor
**Rating:** 4
**Confidence:** 4

**Summary:**

This paper deals with  the 3-DoF camera pose estimation method for a ground-level image with respect to a satellite image. The proposed end-to-end approach that leverages the learning of dense pixel-wise flow fields between the projected ground-level image and the satellite image is built upon RAFT architecture.
Effectiveness of the proposed method is evaluated on competitive benchmarks and outperforms other SoTA methods in both location and azimuth measures.

**Strengths:**

Although the method is simple and the DNN structure is also easily understandable,  the proposed approach has achieved state-of-the-art (SoTA) performance by an adequate margin on various standard benchmarks.
The paper is written clearly and adequate figures support to understand the proposed approach.

**Weaknesses:**

The main difference compared to [29], as mentioned in L. 75-77, is the part where camera poses are estimated from the explicit matching. However, the idea itself is well-known, for example, in [*], so I think the novelty of the approach is limited. While the representation of the loss function may differ, the basic idea has been previously addressed.

[*]A.R. Zamir et al. "Generic 3D Representation via Pose Estimation and Matching", ECCV16.

When dense correspondences are estimated, it is common to use RANSAC-based methods. It would be beneficial to compare the proposed method with RANSAC-based approaches to evaluate their performance.

**Questions:**

- Are the colors in Figure 1 meant to represent the same feature by using the same color? Also, the road appears red in the original image (Figure 1(B)), but it appears green in the BEV (Figure 1(E)). Additionally, why isn't the vertical road in the satellite image shown in red?
- Is Equation 5 correct? When rearranged, it becomes $\frac{1}{1+exp(-\tilde d_i/ \kappa)}$, and $S_i$ disappears.
- Why is the error of Localization/Azimuth for "Ours" in Table 1 (4.97/11.2) different from the values shown in Figure 3 (3.8/8.3)?
-  How is the ground truth for the dense estimation flow in Equation 4 actually obtained? It is mentioned that it is determined by the camera pose in L.188, but in reality, the 3D position (depth) will be required.

**Limitations:**

When it comes to practical use, I believe it is important to verify the robustness and accuracy of the initial value between ground-level images and satellite images.

---

> ### Author Rebuttal · Authors · 2023-08-10
>
> We thank Reviewer-63Cb for the valuable comments. Please find our response below.
>
> **1:Differences compared to Shi and Li[29] and Zamir, Amir R., et al. [\*] are minor.**
>
> From a high-level perspective, Shi and Li [29], Amir R., et al. [\*], and our method all estimate the relative pose from the two-view correspondences. However, our contribution is not a framework that estimates relative camera poses from correspondences. Rather, our contribution is how to estimate correspondences from an extreme stereo pair (ground and satellite images) without ground truth supervision (no 3D data).
>
> Shi and Li [29] assume a fixed correspondence between the ground and satellite view images based on the ground plane homography. However, correspondences computed this way are not accurate for objects above the ground plane. (Please refer to our response to the fifth question of Reviewer-1HCJ for more details.) Amir R., et al. [\*] have ground-truth correspondences between input images for their network supervision, while this is not available in our task.
>
>
> This paper addresses a set of unique challenges in the ground-to-satellite localization task, such as how to train the correspondence estimation network without accurate ground truth supervision, how to select the accurate and robust correspondence pairs from the dense flow map, etc. As a result, our method achieves significantly better performance than the most recent state-of-the-art.
>
>
>
> **2:Comparison with RANSAC.**
>
> Tab.B in the rebuttal PDF presents the comparison between RANSAC and our method. The maximum iteration of the RANSAC is set to 50, and the inlier ratio threshold is 0.5. These parameters are carefully engineered to achieve the best performance. The results in Tab.B show that the performance between RANSAC and our method is similar. Regarding computational speed, the RANSAC takes around 0.3s on average for each image, while our method needs 0.25s.
>
> **3:Visualized colors.**
>
> We used PCA to decrease the high-dimensional feature channels to three channels for RGB visualization. We did not use a fixed PCA for all feature maps. Thus, the same color across different feature map visualizations might be different, and the same object might be visualized with different colors. We will make it consistent in our revision. The vertical road in the satellite map does exist in the feature map, but not obvious. This is because the PCA method only highlights the main features using which the correspondence is correctly estimated. The vertical road is not salient on the ground image; thus, its corresponding features are not highlighted as much as the horizontal road.
>
>
> **4:Mistake in Eq. 5.**
>
> Thank you for pointing out this problem. The correct version should be:
> \begin{equation}
>   \mathcal{L}_c = \sum_{i=1}^n \Vert \frac{S_i}{1+exp(-\tilde{d_i}/\kappa)} + \frac{1 - S_i}{1+exp(\tilde{d_i}/\kappa)} \Vert_1 .
> \end{equation}
> It aims to penalize high confidence for poor matching performance and low confidence for excellent matching performance. The original simple $C_i$ should be changed to $S_i$ to make it consistent with Eq. 3.
>
>
> **5:Inconsistent number between Fig.3 and Tab.3.**
>
> The error of Location/Azimuth for "Ours" in Tab.3 (4.97/11.2) corresponds to the last blue bars (5.0/11.2) in the two subfigures in Fig. 3. The numbers are the performance of our method when trained and evaluated under completely unknown orientation ($\pm 180^\circ$) scenario on the VIGOR dataset.
>
> The numbers (3.8/8.3) in Fig. 3 show the performance of our method when *evaluated* on known orientation ($0^\circ$) but *trained* on completely unknown orientation ($\pm 180^\circ$). This is different from the numbers in fourth row of Tab. 3, because the fourth row of Tab. 3 presents the performance of our method when trained and evaluated on the known orientation ($0^\circ$) scenario.
>
> We will make this clearer in the paper.
>
>
> **6:How to obtain ground truth for dense estimation flow.**
>
> The ground truth optical flow between the ground and satellite image is hard to be derived, given the absence of 3D data. Thus, we propose to convert the ground image to the Bird's-eye view (BEV) and estimate the correspondences between the BEV image (/feature map) and the satellite image (/feature map). In this case, there is a inplane rotation and translation difference between the BEV image and the satellite image. Thus, their pixel correspondences can be computed by a homography derived from this relative rotation and translation. This relative rotation and translation is also the ground truth relative pose between the ground image and the satellite image.
>
>
> **7:Robustness to different initial values.**
>
> The location and orientation search range used in the experiments in the paper follow the official settings in their original papers [6, 14, 23].
>
> The coarse locations of ground cameras on the VIGOR dataset are in the center quarter of the satellite image. The search range is approximately $36\times 36$ m$^2$. The location search range for the KITTI, Ford multi-AV, and Oxford RobotCar datasets is about $40 \times 40$ m$^2$. The error of the coarse pose in the experiment settings is similar to a consumer-level noisy GPS.
>
> We further present the performance of our method across a broader location search range in Tab. C of the rebuttal PDF, specifically, $60 \times 60$ m$^2$. As the location search range increases, so does the location ambiguity, leading to a slight decline in our method's performance. However, we should note that even with the larger location search range of $60 \times 60$ m$^2$, our approach outperforms the recent state-of-the-art SliceMatch [6] when it is tested within a smaller location search range of $40 \times 40$ m$^2$.
>
>
> The experiments on varying orientation ambiguity are presented in Fig. 3 of the paper.

---

> > ### Comment · Reviewer_63Cb · 2023-08-13
> >
> > Thank you for answering my concerns. Some of the concerns are resolved. Please find the following comments.
> >
> > > 2. Comparison with RANSAC
> >
> > From the results in Tab.B, the advantage of the least squares is marginal with respect to accuracy and computation speed.
> > Althogh the authors claim the least squares pose regression module is one of the proposal in L.117, I belive its effectiveness is limited.
> >
> > >3. Visualaizated colors
> >
> > Is it possible to show examples of correctly/incorrectly matched feature points overlayed on both ground and satellite images? As the color aligned feature maps were not provided in rebuttal and the correspondences in Fig.4 are small, I would like to know the qualitative results.

---

> > > ### Author Response · Authors · 2023-08-16
> > > **RE: Official Comment by Reviewer 63Cb**
> > >
> > > We thank Reviewer 63Cb very much for the further response. Please find our response below, and let us know if there is anything unclear.
> > >
> > >
> > > >>2. Comparison with RANSAC
> > > >
> > > >From the results in Tab.B, the advantage of the least squares is marginal with respect to accuracy and computation speed. Althogh the authors claim the least squares pose regression module is one of the proposal in L.117, I believe its effectiveness is limited.
> > >
> > > Yes, during evaluation, the effectiveness of the least squares pose regression over RANSAC is limited. Our original intention in L117 of the paper is to say that differentiable least square pose regression enables training of the optical flow network through pose loss（Eq. 6). We will make this clearer.
> > >
> > > >> 3. Visualized colors
> > > >
> > > >Is it possible to show examples of correctly/incorrectly matched feature points overlayed on both ground and satellite images? As the color aligned feature maps were not provided in rebuttal and the correspondences in Fig.4 are small, I would like to know the qualitative results.
> > >
> > > Sure. We provide modified visualizations in this anonymized link  (https://drive.google.com/file/d/1HF0tc628X8nGc899_yZX5mD33HFP_0iA/view?usp=drive_link).
> > >
> > > To ensure consistent colors, we use the same PCA compression for the BEV and satellite feature maps. We differentiate between  high-confidence and low-confidence matches using a threshold of 0.5. We also show the estimated corresponding points on the original ground and satellite images for clarity.  In these visualizations, circles indicate higher-confidence matching points, while stars represent lower-confidence ones.
> > >
> > > From the visualizations, it can be seen that the confidence for pixels in the BEV image that are co-visible in both ground and satellite images (e.g., road area) tends to be high, while the confidence for pixels that are only visible in one view (e.g., building roof, tree canopy, occluded area) tends to be low. As the result, our confidence map effectively select reliable matching points for accurate pose estimation.

---

### Official Review · Reviewer_1HCJ · 2023-07-04

**Soundness:** 2 fair
**Presentation:** 2 fair
**Contribution:** 2 fair
**Rating:** 3
**Confidence:** 5

**Summary:**

In this paper, the authors try to address the cross-view geo-localization by learning dense pixel-wise flow fields between ground and satellite images. To achieve this, they utilize a ground feature projection module to align the spatial feature and propose a refinement block to enhance the context feature for accurate optical flow estimation. However, some details about refinement modules, data collection, and experiment setup are unclear.

**Strengths:**

This paper tries to address the cross-view geo-localization by learning dense pixel-wise flow estimation. And the authors conduct an account of experiments on KITTI, Ford multi-AV, VIGOR, and Oxford RobotCar.

**Weaknesses:**

However, there are some concerns about this paper:
1. More detail about the satellite image collection. In Sec 4.1, the authors only present the detail about the ground-view dataset. Is there any detail about the satellite image, including collection, preprocessing, ground sample distance, data distribution, and so on?
2. As stated in Line 146, RefineBlock is a crucial module of the proposed method. However, the authors provide no detail about RefineBlock. Please provide the detail about RefineBlock. And more analysis and experiments are appreciated to demonstrate the importance of RefineBlock.
3. LM [1] are trained with 2 epochs. However, this method is needed to train with 25 epochs in Line 237, which is 10x longer than LM [1]. Are the results of LM [1] in Table 1 and Table 2 trained with 25 epochs? Could you provide the results of LM [1] trained with 25 epochs?
4. The training schedule is unclear. In Line 235, the feature extractors are optimized with 12 epochs in the Dense Flow Estimation stage. However, in Line 237, the pipeline is conducted with 25 epochs in an end-to-end schedule. And in Line 207, the momentum parameter is changed to 10 after 15 epochs. Hence, I am confused about the training schedule. Does it mean the whole pipeline consists of two stages and is trained with 37 epochs in total?
5. The most similar state-of-the-art work is proposed by Shi and Li [1]. In Table 4 of LM [1], the results show that LM achieves better translation performance but inferior rotation performance than the network-based optimizer, including GRU. Hence, the relationship and main difference with the prior work [1] are appreciated to be discussed in the related work. It is better to analyze the advantages and disadvantages of the proposed approach compared with LM [1]. And more experiments and analyses are appreciated.

Reference

[1] Yujiao Shi and Hongdong Li. Beyond cross-view image retrieval: Highly accurate vehicle localization using satellite image. CVPR 2022.

**Questions:**

See the Weaknesses.

**Limitations:**

The limitations and potential negative societal impact are not explicitly discussed.

---

> ### Author Rebuttal · Authors · 2023-08-10
>
>
> We thank Reviewer-1HCJ for the valuable comments. Please find our response below.
>
> **1: Details of satellite images in the datasets, including collection, preprocessing, ground sample distance, data distribution, and so on.**
>
> Please refer to our response to all the reviewers in the overall response.
>
>
> **2:Explanation of the RefineBlock.**
>
> The RefineBlock is proposed to address the limitations of the ground-plane homography-based ground-to-overhead projection, including incorrect correspondences for objects above the ground plane, distortions in the projection, etc. In order to eliminate these issues, the network should have a larger receptive field of the input overhead-view feature map projected based on the ground plane homography to achieve a better and high-level understanding of the feature map. Thus, a 7x7 convolutional layer is first employed in the RefineBlock. Then, a 3×3 residual block is applied to refine the details in local regions. Finally, a 1×1 convolutional layer is adopted to facilitate feature interaction and integration, introducing enhanced non-linear transformation capabilities in the feature space. Our source code containing all implementation details will be released.
>
>
> **3：The results of longer training epochs of LM [29].**
>
> Thank you for the suggestion. We indeed trained the LM method for longer epochs when preparing this submission, but we did not observe too much performance increase with longer training epochs. Please refer to Fig. A in the rebuttal PDF for the results. Thus, we choose to use the released performance in the original paper for comparison. We would like to modify the numbers to our reproduced results with a longer training epoch (25) for fair comparison.
>
> We should also note that our method has a significantly faster training and evaluation speed than the LM method. For example, on the same NVIDIA TITAN V GPU with 12GB memory, our method takes 2 hours for training one epoch, while LM needs 6 hours; our method achieves a 0.25s evaluation time on average for each ground image, while LM needs 2s for one ground image's evaluation.
>
>
>
> **4:Unclear training schedule.**
>
> Thanks for pointing out this confusing illustration. In line 235, the optimization of the Dense Flow Estimation stage refers to iterating the GRU structure 12 times, independent of the training process. The entire training schedule involves 25 epochs. At the 15th (out of 25) epoch, we change the value of $\beta$ to 10.
>
>
> **5:Differences from LM.**
>
> The differences between LM and our method include different network architectures, different projection directions for bridging the domain gap (satellite-to-ground Vs. ground-to-satellite), different optimizers (LM Vs. GRU), different optimization parameters (3-DoF pose Vs. dense flow map) and different training objective functions.
>
>
> From a high-level perspective, both the LM paper and our method estimate the relative pose from the two-view correspondences.
> The LM paper aims to optimize the relative pose by minimizing the feature differences of corresponding pixels in the two-view images.
> However, the correspondences of pixels are computed based on ground plane homography (and the relative camera pose), which is incorrect for objects above the ground plane.
> Minimizing the feature differences between non-corresponding features would lead to sub-optimal relative poses.
>
> Thus, in this paper, we aim to estimate accurate correspondences between the two views first and compute the relative pose second.
> The correspondences are estimated in a way that is agnostic to relative poses.
> Once the correspondences are accurate and robust enough, the relative pose can be directly computed by solving a least squares problem instead of training an additional network.
> Thus, the main capacity of our method is spent on accurate correspondence learning instead of the relative pose, while the LM paper spent its main effort on pose estimation by a fixed correspondence relationship according to the pose.
>
>
> To achieve this goal, our paper addresses a set of unique challenges encountered, such as how to train the correspondence estimation network without accurate ground truth supervision, how to select the accurate and robust correspondence pairs from the dense flow map, etc. As a result, our method achieves significantly better performance and a faster evaluation speed than the LM method.

---

> > ### Comment · Reviewer_1HCJ · 2023-08-19
> >
> > Thanks so much for the detailed response. But the presentation of this manuscript is not good enough. Most of the key details are not introduced in the paper. The RefineBlock looks simple and limited novelty. And what problems existed in current LM and other GRU methods have been solved in this paper? I am inclined to keep my rating for now.

---

> > > ### Author Response · Authors · 2023-08-20
> > >
> > > Thank you very much for the further response.
> > >
> > > Regarding the question **"what problems existed in current LM and other GRU methods have been solved in this paper** ", our response is as below.
> > >
> > > The LM paper assumes a ground plane homography for the ground-and-satellite image correspondences. This assumption, however, is *inaccurate for scene objects above the ground plane*. Thus, in this paper, we focus on estimating *more accurate correspondences between the two views*. Only when the correspondences are accurate enough, can the camera pose correctly estimated from the correspondences.
> > >
> > > Compared to GRU methods for optical flow estimation, this paper focuses on a challenging scenario where the two input images suffer from significant viewpoint change and, more importantly, no ground truth is available for supervision.
> > >
> > > Specifically, this paper discovers that the supervision is computable when converting the ground view to the overhead view by a BEV-synthesis, and introduces additional implicit supervisions either to strengthen the supervision (Eq. 6) or to filter out unreliable correspondences (Eq. 5.). Our contribution is not simply applying existing new techniques to a new task. As demonstrated in Tab. 5, this does not achieve desirable performance. Rather, only when incorporating our proposed solutions for addressing the unique challenges, as described above, can the method achieve the best performance.
> > >
> > > Please let us know if there is anything unclear.

---

### Official Review · Reviewer_Bgje · 2023-07-05

**Soundness:** 3 good
**Presentation:** 3 good
**Contribution:** 2 fair
**Rating:** 5
**Confidence:** 4

**Summary:**

The paper presents a method that estimates a 3-DoF pose that aligns correspondences between ground-level and satellite/aerial images. The proposed solution uses a CNN-based model to extract image features, map image features from the ground-level camera to a BEV so that it can be matched with features from the satellite image. Lastly, a least-square regression us run to estimate the final 3-DoF pose (rotation and translation). The paper presents results on the KITTI and Ford multi-AV datasets where the proposed method consistently shows performance improvements over a few baselines.

**Strengths:**

In sum, I think the proposed solution is simple and straightforward. Thus, I think the proposed approach is easily reproducible.
The particular strengths I see are the following:

S1. Simplicity and straightforward of solution. The solution essentially mimics a classical dense matching of features coming from two different images (i.e., aerial/satellite view and ground-level view). Because of this simplicity, I think the proposed solution can be easily adopted and reproduced.

S2. End-to-End approach. I do like that the proposed solution includes the feature matching pipeline but also optimizing for the 3-DoF pose which ultimately is the final solution. This is great as it directly learns features that are useful to obtain the desired pose.

S3. Use of a confidence value as part of the solution is also very useful. I think this should be able to handle outliers and allow a robust computation of the estimated pose.

**Weaknesses:**

While I value the simplicity of the solution, I question its novelty. To me, the novelty is very important as this is the way we break common perspectives and open new avenues of research. In detail here are my concerns about novelty:

W1. The proposed solution uses CNN features in the era of Transformers. While this is a feasible solution (as shown by the experiments), it has been shown in the computer vision community that attention mechanisms can help in feature matching and also deal with outliers; a good example of this is LoFTR presented at CVPR 2021. Unfortunately, I don't see any justification in the paper about the use of CNNs when Transformers inherently provides mechanisms to do feature matching.

W2. The proposed solution mainly is composed of well known components, and the proposed method is quite straightforward as it follows classical pipelines of feature matching (e.g., PatchMatch-Based Neighborhood Consensus for Semantic Correspondence CVPR 2021) combined with well known optimization methods to estimate a pose.

W3. The paper uses terms in a wrong way. For example, in Eq. 3 the $\mathbf{t}$ is introduced as position in line 119. However, the Eq. 3 shows this is a translation and not a position. This is very confusing as it conflicts the math and the stated definition, which decreases clarity. Second, the $G(\cdot)$ introduced in line 141 is not defined. What is it? Lastly, in Eq. 3, the loss seems to be using L2 norm and not using a squared error on scalars as suggested by the parenthesis in the equation. Thus, the "Position Loss" described in 199 is not optimizing directly the position but rather the pose since it penalizes deviations of the rotation angles and translations.

--------
Post Rebuttal.

I am satisfied with the answers the authors have provided. Although I think the proposed is simple and straightforward, I think it produces good results according to the experiments. I think the community can get intrigued by the results and the simplicity of the methods, and thus I expect this paper to cause some interest in the CV community. For these reasons, I will upgrade my rating to borderline accept.

**Questions:**

Please see Weaknesses section.

**Limitations:**

The limitations of the paper are not properly discussed. I think the paper may suffer from images with repetitive features (e.g., an image of a large field of grass, forests, etc.). It is unclear if the proposed method of estimating a confidence can handle this cases correctly. Lastly, since this technology matches aerial views with ground-level views, this can be used for surveillance. I think this an ethics review for this case.

---

> ### Author Rebuttal · Authors · 2023-08-10
>
>
> We thank Reviewer-Bgje for the valuable comments. Please find our response below.
>
> **1:Reasons for using CNN.**
>
> There are currently two lines of work for estimating correspondences between two images (or two point clouds, etc.) The first line of work first constructs a noisy cost volume between the two images using the features extracted by a simple network (usually CNNs), then refine the noisy cost volume to a clean one by an additional network (3D CNN, RNN, etc.).
> The other line of work aims to learn strong and discriminative representations by self and cross attentions (transformers) for each pixel in the image and then compute the cost volume (or similarity matrix), from which the correspondence is directly computed by an argmin (or argmax) operation.
> Our method of using RAFT aligns with the first category, and the LoFTR belongs to the second category.
>
> Determining which method is generally better is hard, especially when ground truth correspondence supervision is available. While in our task where such ground truth correspondence supervision is not available, we empirically find that a simpler network architecture is easier to learn.
>
> Specifically, we aim to estimate the correspondences between the ground and satellite images to estimate their relative pose. However, no ground truth correspondence is available for these two view images to supervise a network. Thus, we propose to first convert the ground view image to a Bird's-eye view (BEV) and then estimate the correspondences between the BEV image and the reference satellite image. In this case, the GT correspondences between the BEV and the satellite image can be computed from their GT relative pose. However, due to the severe occlusions between the ground and overhead view images, it is impossible to synthesize a perfect BEV image corresponding to the ground image. Thus, the computed GT correspondences are not really applicable to every pixel in the BEV image. A learnable confidence map might solve this issue, by filtering out unreliable correspondences. However, the confidence map does not have GT supervision either.
> We further propose a new mechanism to supervise this confidence map. One is from the estimated flow, indicated in Eq. 5. The other is applying the discrepancy between the computed pose from the estimated flow map and the GT pose to supervise the confidence map and estimated optical flow map implicitly.
>
>
> In summary, we have explored various signals to supervise the optical flow (correspondence) estimation network, but all these signals are implicit and not ground truth. Given such weak supervision, it is hard to train a complex network built by transformers, and we indeed observe the non-convergence phenomenon when using LoFTR for correspondence estimation in our task.
>
>
> For comparison in terms of time complexity and memory consumption, employing the LoFTR pipeline for inference requires approximately 400ms and consumes 10GB of storage, whereas utilizing GRU for inference only necessitates 200ms and occupies 6GB of storage. The experimental results of this comparison are presented in Tab.A of the rebuttal PDF.
>
>
> **2:Novelty.**
>
> We acknowledge that first conducting feature matching to establish correspondences and then computing the relative pose from the correspondences is a common structure, and our method aligns with this pipeline. However, we want to highlight that our contribution is not a simple adaption of this common pipeline to a new task, but how to address the unique challenges in the new task faced by this existing technique.
>
> Specifically, the GT correspondences between the ground and satellite image in our task, neither pixel-wise nor keypoint-wise, is unknown, given the absence of the 3D data. Furthermore, due to the extreme viewpoint change between the ground and satellite images and their different capture time, it is hard to directly use a network pre-trained on existing data with GT correspondences (usually both ground views and captured at a close time) to solve our task. Thus, our main contribution is how to estimate correspondences between the extreme stereo pair without GT supervision and then use the estimated correspondences for relative pose estimation.
>
> This paper discovers that the supervision is computable when converting the ground view to the overhead view by a BEV-synthesis, and introduces additional implicit supervisions either to strengthen the supervision (Eq. 6) or to filter out unreliable correspondences (Eq. 5.). Please refer to our response to the first question for more details. The experiments in Tab. 5 also demonstrate that naively applying existing techniques for solving the task achieves undesirable performance. Only when incorporating our proposed solutions for addressing the unique challenges can the method achieve the best performance. Our improvement over the state-of-the-art is also significant.
>
> We believe this paper should also benefit some similar tasks. For example, the self-supervision in Eq. 5 for the confidence map also applies to other self-supervised optical flow estimation tasks.
>
> **3:Error in variable $\boldsymbol{t}$ in Eq. 3.**
>
> The variable $\mathbf{t}$ in L119 denotes the relative position of the ground camera with respect to the reference satellite image center. Thus it is also the relative translation between the ground camera coordinate system to the satellite camera coordinate system in Eq. 3.
>
>
> **4:Meaning of variable $G(.)$ in line 141.**
>
> $G$ represents the process of projecting point $\boldsymbol{p}$ onto the Bird's Eye View (BEV) to obtain point $\boldsymbol{p}'$. We will make this clearer in the paper.
>
> **5:Ambiguity in the position loss at line 199.**
>
> Thank you for pointing out this issue. The "Position Loss" is actually the "Pose Loss" as the reviewer mentioned. We will correct this issue in the revision if accepted.

---

> > ### Comment · Reviewer_Bgje · 2023-08-12
> > **RE: Rebuttal by Authors**
> >
> > Thanks for the clarification on my points. Although I think the proposed method is simple and straightforward, I see value in the proposed approach as it obtains good localization results as shown in Tables 3 and 4 and the supplemental material. Please consider the inclusion of clarifications made to other reviewers if the paper ends up being accepted.

---

> > > ### Author Response · Authors · 2023-08-20
> > >
> > > Thank you very much for the support.
> > >
> > > We promise all the clarifications and suggestions made by all the reviewers will be included in the paper.

---

### Official Review · Reviewer_pNph · 2023-07-06

**Soundness:** 4 excellent
**Presentation:** 4 excellent
**Contribution:** 4 excellent
**Rating:** 8
**Confidence:** 3

**Summary:**

The paper introduces a method for estimating 3DOF camera pose of satellite images given a ground-level image and an initial pose estimate as input. The ground-level image is mapped to a satellite view using the camera parameters, confidence-weighted 2D matches are estimated between the warped image and the satellite image, from which the 3DOF parameters are solved. The authors show that this approach works well in the wild across many standard datasets, and outperforms prior work by a large margin.

**Strengths:**

The method follows a procedure that has worked very well for other 3D vision problems (SLAM, SfM): 1) Align features 2) 2D-2D matching 3) Solve for camera pose which explains the 2D observations. This sounds more principled than the prior work.

The method is interpretable and learns outlier rejection automatically.

The paper is very well written and easy to understand.

The authors provide an analytical solution to their solver layer, as opposed to prior work which uses the LM algorithm to align features.

Strong empirical results across many datasets compared to prior work.

Paper includes ablation experiments on several of the components

**Weaknesses:**

The related work section doesn’t explain the main difference between the proposed method and the referenced methods. For example, what is the main difference between *Wang et al.[42]* and the proposed method? At face-value they sounds similar.

Paper is missing resource requirements (time or memory) compared to prior work

**Questions:**

Why do the confidence estimates need their own loss? Can they not be learned automatically just by supervising on the solver output (e.g. DROID-SLAM)?

In Eq. 1, how is T obtained? Is it provided as input or is there a canonical ground-camera pose assumed (e.g. upright, pointing north, 0 translation)?

**Limitations:**

No limitations section, so this is not clear.

---

> ### Author Rebuttal · Authors · 2023-08-10
>
> We thank Reviewer-pNph for the valuable comments. Please find our response below.
>
> **1:Missing resource requirements.**
>
> Our method is trained and evaluated with an NVIDIA TITAN V GPU with 12GB memory. The average evaluation time for each ground image is 0.25s. Using the same GPU device, the evaluation time of the LM method [29] (CVPR 2022) is 2s, which is around eight times longer than ours. The code and trained models of SliceMatch have not been released yet. Thus, we cannot compare with it now, but we would like to add it when their code is publically available.
>
>
> **2:Difference with Wang et al.[42].**
>
> Wang et al. [42] require 3D LiDAR points for localizing a ground image, while our method does not rely on any 3D points. Thus, our method tackles a more challenging scenario.
>
> Compared to other works on ground-to-satellite image localization, this paper is the first to demonstrate the possibility of estimating the optical flow between the ground-and-satellite image pair with extreme viewpoint change for relative pose estimation without the ground truth optical flow map as supervision.
>
>
>
>
> **3:Necessity of the confidence loss.**
>
> The confidence map selects "good correspondences" between the synthesized BEV map and the reference satellite image that contributes an accurate relative pose. Only applying supervision on the solver output is able to provide implicit supervision signals on the confidence map. The results of this ablation variant are shown in the second last row of Tab. 5. However, this supervision might be weak. Using the additional independent supervision proposed by this paper, we can further strengthen the supervision signal and enhance the pose estimation accuracy, as demonstrated by the last row of Tab. 5.
>
>
>
> **4:Meaning of variable $\mathbf{T}$ in Eq. 1.**
>
> Here, we put the origin of the world coordinate system ($x^W$) on the ground, with the directions of the three axes parallel to the three axes of the camera coordinate system. Thus, there is no rotation difference but a translation difference between the world and camera coordinate systems. The translation difference is along the vertical direction, because the origin of the camera coordinate system is above the ground plane. We use $\mathbf{T}$ to denote this relative translation and omit the rotation matrix when writing Eq. 1. We will clarify this in the paper.

---

> > ### Comment · Reviewer_pNph · 2023-08-16
> >
> > Thank you for your response. All my questions and concerns have been addressed.

---

### Author Rebuttal · Authors · 2023-08-10

We appreciate all reviewers' valuable feedback and constructive suggestions. We are glad that two ethics reviewers confirmed that the datasets used in this paper are all licensed for academic use and not original to this work. We are committed to adhering to the guidelines set forth by the original data publishers. Our approach will remain adaptable and responsive to any modifications in these guidelines.

Both ethics reviewers asked for details about the datasets. Reviewer-1HCJ also inquired about the details of satellite images in the datasets, including the collection, preprocessing, ground sample distance, data distribution, etc. Our response is as follows. Point-to-point responses for other questions are provided for each reviewer, respectively.

**1:More detailed information about the datasets.**

(1) The satellite images of the **KITTI** and **Ford Multi-AV** datasets were collected by Shi and Li [29].
The ground images are from the raw data for the cross-view KITTI dataset. Each ground image is accompanied by a satellite image whose center corresponds to the location of the ground camera. This dataset is divided into Training, Test1, and Test2. Images in Test1 come from the same region as those in the training set, while images from Test2 are from other regions. For the Ford Multi-AV dataset, we exclusively utilize data from V2 vehicles. The authors first drew a trajectory of the ground view images and then collected satellite images along the trajectory. Ground images captured on one date are used for training, and those captured on another date are used for testing. Satellite images are kept the same during training and testing. For implementation in both datasets, the corresponding satellite image for each ground image is randomly rotated and translated within an error range to mimic the misalignment between the two view images, and research is conducted to estimate the relative pose between the two images. The ground resolution of the satellite images in these two datasets is around 0.2m per pixel, and the coverage of the satellite images used is around 100m x 100m.

(2) The satellite images of the **VIGOR** dataset were collected by Zhu et al. [14]. The ground images are north-aligned panoramas, and they are from four cities:  New York, San Francisco, Chicago, and Seattle. There is a positive satellite image in the database for each ground image, meaning that the ground image is within the center 1/4 quarter of the satellite image. All the satellite images are north-aligned. The dataset includes two evaluation splits: same-area and cross-area, based on whether the images in training and testing sets are from the same region. Research is conducted to estimate the relative pose between the ground image and its positive satellite image center. In this dataset, the ground resolution of satellite images for New York, San Francisco, Chicago, and Seattle is 0.113, 0.118, 0.111, and 0.101, respectively. The size of the aerial views for these four cities is 640×640.

(3) The satellite images of the **Oxford RobotCar** dataset were collected by Xia et al. [23]. They provided a very large satellite map covering the whole region where the ground images were collected. The ground images were collected through multiple traversals on a single route in Oxford, UK, encompassing various time periods, seasons, and weather conditions. During implementation, for each ground image, a random relative translation is generated, and a small satellite patch corresponding to the randomly generated pose is extracted from the large satellite map. Research is conducted to estimate the relative pose. The ground sample distance of the satellite images in this dataset is 0.0924m per pixel, and the coverage of the satellite images used is around 55m x 55m.

**2:Is the dataset being used in its entirety?**

The KITTI, VIGOR, Ford multi-AV, and Oxford RobotCar datasets we used are approximately 409GB, 183GB, 361GB, and 499GB, respectively. After downloading these datasets, we store them on a NAS hard drive.

When using the KITTI [9] and Ford multi-AV datasets [2,29], we follow the setup described by Shi and Li [29]. For the KITTI dataset, daytime images captured from different trajectories at various times are used. Regarding the Ford multi-AV dataset, images captured by the front left camera of the V2 vehicle are used. For the VIGOR dataset, we follow SliceMatch [6] in using the ground and its positive satellite images. The semi-positive satellite images are not used.

Xia et al. [43] selected a portion of the whole Oxford RobotCar dataset and supplied it with a large satellite map. Specifically, they selected several traversals recorded on different days and at various times of the day to capture different weather conditions. The weather conditions were labeled as "sunny," "cloudy," or "overcast." Additionally, the traversals were subsampled to ensure a minimum distance of 5 meters between consecutive frames. In the end, the dataset consisted of 17,067 images for the training set, 1,698 images for the validation set, and 5,089 images for the test set, all of which are ground-level frontal views. Our usage of this dataset follows this setup.


**3:Computational cost.**

We utilize an NVIDIA TITAN V GPU with 12GB memory for training and testing. Our method takes two hours for training one epoch, 0.25s on average for estimating one ground image's pose during evaluation. During the evaluation, our method consumes 6GB GPU memory for one ground image.

**4:Can the code be shared?**

The source code is provided in the supplementary material. We will make it publically available to facilitate reproducible research.

---

### Decision · Program_Chairs · 2023-09-21

**Decision:**

Accept (poster)

**Comment:**

Most of the reviewers have given it a positive rating, with one reviewer championing for its acceptance, but not all the positive reviewers were enthusiastic with the draft.

Meta-reviewer do agree with 1HCJ that authors should clarify and detail all the points regarding architecture, training, experiments etc...  At the same time  Meta-reviewer agree with reviewer pNph that "fact that the overall approach is simple but works very well is a virtue, not a detriment". This however, should be the duty of authors to make such case by improving their presentation and writing of the paper. After a discussion with SAC it was decided that it should be accepted to NeurIPS. We hope that authors will update the paper to include information about RefineBlock, clarify difference between their work and existing state-of-the-art  (especially Wang et al. [42] and [29]) in the final submission, and add other clarifications requested by the reviewers.
 Overall draft itself need careful proofreading to improve clarity.